# Analysis of the Challenges of Artificial Intelligence of Things (AIoT) for the Smart Supply Chain (Case Study: FMCG Industries)

**DOI:** 10.3390/s22082931

**Published:** 2022-04-11

**Authors:** Hamed Nozari, Agnieszka Szmelter-Jarosz, Javid Ghahremani-Nahr

**Affiliations:** 1Department of Industrial Engineering of Central Tehran Branch, Islamic Azad University, Tehran 1469669191, Iran; ham.nozari.eng@iauctb.ac.ir; 2Department of Logistics, Faculty of Economics, University of Gdańsk, ul. Armii Krajowej 119/121, 81-824 Sopot, Poland; 3Faculty Member of ACECR, Development and Planning Institute, Tabriz 5154837693, Iran; javid.ghahremani@yahoo.com

**Keywords:** supply chain management, Artificial Intelligence of Things, smart supply chain, digital systems, logistics, nonlinear prioritization

## Abstract

In today’s competitive world, supply chain management is one of the fundamental issues facing businesses that affects all an organization’s activities to produce products and provide services needed by customers. The technological revolution in supply chain logistics is experiencing a significant wave of new innovations and challenges. Despite the current fast digital technologies, customers expect the ordering and delivery process to be faster, and as a result, this has made it easier and more efficient for organizations looking to implement new technologies. “Artificial Intelligence of Things (AIoT)”, which means using the Internet of Things to perform intelligent tasks with the help of artificial intelligence integration, is one of these expected innovations that can turn a complex supply chain into an integrated process. AIoT innovations such as data sensors and RFID (radio detection technology), with the power of artificial intelligence analysis, provide information to implement features such as tracking and instant alerts to improve decision making. Such data can become vital information to help improve operations and tasks. However, the same evolving technology with the presence of the Internet and the huge amount of data can pose many challenges for the supply chain and the factors involved. In this study, by conducting a literature review and interviewing experts active in FMCG industries as an available case study, the most important challenges facing the AIoT-powered supply chain were extracted. By examining these challenges using nonlinear quantitative analysis, the importance of these challenges was examined and their causal relationships were identified. The results showed that cybersecurity and a lack of proper infrastructure are the most important challenges facing the AIoT-based supply chain.

## 1. Introduction

Over the past decade, supply chain management and information technology management have attracted much attention from practitioners and researchers. As can be deduced from information technology, companies have a strong tendency towards integration; integration is therefore considered an effective technique in the supply chain by sharing effective information to improve the performance of the sensitive supply chain. Thus, the growing interest in supply chain management has been driven by competitive pressure and has led to its eventual promotion to become an essential part of corporate operations and strategy [1]. On the other hand, due to the high speed of changes in the market and customers, companies need to continuously improve their supply chain management systems to provide the right product to the right customer and at the right time. The integration of new technologies in the supply chain has made it possible to improve the exchange of information and facilitate the monitoring of physical goods throughout the supply chain. Information technology developments play an essential role in increasing the planning, implementation and flow control, and storage of goods, services, and information from the point of origin to the point of consumption to increase customer satisfaction. The Internet of Things (IoT) is one of the latest developments in information technology and a new revolution in this field, which has provided a paradigm shift in various fields, including supply chain management. When artificial intelligence (AI) is added to IoT, it means that devices can analyze data, make decisions, and act on these data without human intervention [2]. The complete combination of the IoT with artificial intelligence, known as AIoT, allows companies to take advantage of both at the same time. This technology’s capabilities, such as transparency, agility, and adaptability to the supply chain, offer tremendous opportunities to address supply chain management challenges more effectively. As smarter, more technology-driven, and more intertwined supply chains grow, research into the IoT and its innovative applications in supply chain management is growing faster. There are several devices in IoT and artificial intelligence in the supply chain [3], such as autonomous and self-driving vehicles, warning sensors, and customer information. However, the most interesting use of the IoT and artificial intelligence is to embed intelligent sensors in a product’s packaging and analyze their data to allow customers to track their goods across the supply chain until the final delivery stage. Of course, due to the presence of Internet and network technology and analyses based on them, and the complexity of using these technologies, it can be realized that the simultaneous presence of these technologies can bring challenges [4].

Given the growing use of IoT and artificial intelligence in business areas, it seems that examining the dimensions and challenges of implementing an intelligent supply chain that uses these technologies simultaneously can be of great importance. Therefore, a proper understanding of the key elements and components affecting a supply chain based on the integrated use of these technologies seems essential. In this regard, in this paper, an attempt has been made to examine the effective parameters of these technologies on the supply chain to extract the most important challenges facing them. In this regard, fast-moving consumer goods (FMCG) companies have been selected as the case study due to the availability of specialists. Moreover, FMCG are products that are sold quickly and at a relatively low cost. Using quantitative analysis of experts’ opinions in this field, the effectiveness of these supply chain implementation challenges based on the Artificial Intelligence of Things and its internal effects were determined. Therefore, using the results of this paper, we can try to solve the most important challenges in the field of the simultaneous use of IoT technologies and artificial intelligence to increase the efficiency of the supply chain [5].

The organization of this paper is as follows. In Section 2, the theoretical foundations and review of the research background are discussed. In Section 3, the most important challenges facing an intelligent supply chain using the convergence of artificial intelligence and the IoT are identified based on a systematic review of the literature as well as interviews with supply chain and information technology (IT) activists in FMCG companies. In Section 4, the research method, which includes the pattern of causal relationships between the studied variables using the DEMATEL decision-making method, as well as the nonlinear prioritization method based on hierarchical analysis, is presented. Section 5 presents the results and findings. Finally, the conclusions, future research opportunities, and limitations of this research are presented in Section 6.

## 2. Literature Review

This section presents brief information about the history and definition of Artificial Intelligence of Things. Moreover, in addition to expressing the role of AIoT in supply chain management, the research background is reviewed.

### 2.1. Artificial Intelligence of Things

The IoT and artificial intelligence are powerful technologies. When AI and IoT are combined, Artificial Intelligence of Things or AIoT is obtained. In other words, the IoT is a digital nervous system, and artificial intelligence is the brain of this system. AIoT is revolutionary for both types of technology and beneficial for both types of technologies because artificial intelligence values the IoT through machine learning capabilities and IoT values artificial intelligence through connectivity, signaling, and data exchange. As IoT networks are spread across large industries, there will be large amounts of human-centered, machine-generated data [6]. AIoT can support data analysis solutions that can add value to the data generated by the IoT. Although some IoT systems are designed for simple event control in which the sensor signal generates a similar response, such as turning on/off a light based on changes in ambient light, many events are much more complex and can be used to interpret events using analytical techniques. AIoT is in place to take the proper steps to achieve this. It place smart tools on edge and gives devices the ability to perceive data, observe their surroundings, and make the best decisions—all of which can be done with minimal human intervention. With the power of artificial intelligence, AIoT devices are not merely messengers that deliver information to the control center, but have become intelligent machines capable of self-centered analysis and operating independently [7].

In terms of data analysis, AIoT technology combines machine learning with IoT networks and systems to create “learning machines”. This can then be applied to enterprise and industrial data to control IoT data, such as network edges, to automate tasks in a connected workplace. Real-time data are of key value for all AIoT applications and solutions. In a specific usage example, AIoT solutions can also be integrated with social media and HR platforms to create an AI decision as a service function for HR professionals. There are four main areas that AIoT affects: wearables, smart home, smart city, and smart industry. Some research has been done in these key areas. In a study, Xiong et al. [7] proposed an AIoT-based system for the real-time monitoring of tunnel construction. They showed that the AIoT-based system improves information and automation during construction, facilitates decision-making, and prevents accidents. Guo et al. [6] created a prior-dependent graph for data clustering and dimension reduction at the AIoT edge. In their research, Sun et al. [8] presented an energy-efficient and fast design for hybrid storage-class memory in an AIoT terminal system. They showed that the proposed system could reduce energy consumption by an average of 46.2% compared to the traditional system. Sun et al. [8] examined the virtual store of AI-enabled objects (AIoT) using an automated sensor-enhanced robotic software manipulator. Their findings showed that using IoT analysis and artificial intelligence (AI), a virtual store can provide customers with real-time feedback on product details. Chen [9], in a paper examining the implementation of an intelligent linking service on AIoT, examined the hierarchy for material flow management. Hu et al. [10] proposed a new two-step unsupervised error detection framework combining feature extraction and fuzzy clustering for common AIoT. In a paper, Yang et al. [11] used industrial AIoT to improve quality at an HP plant. The results showed the high impact of this technology on quality improvement.

### 2.2. AI and IoT-Based Supply Chain

Many businesses have embraced artificial intelligence and the IoT as part of their processes and products in today’s world. Recent studies have shown that these are two well-known technologies that are used seamlessly today. They were also found to be the best technologies that companies invest in to increase efficiency and create a competitive advantage [12]. Thus far, several definitions have been proposed for the IoT. However, the definition provided by Ben-Daya et al. [13] emphasizes the concept of the IoT and supply chain management and, therefore, has been considered in this paper. Based on this, the IoT can be defined as follows: the Internet of Things is a network of physical objects that are digitally connected to sense, monitor, and interact within a company and between the company and its supply chain, enabling agility, visibility, tracking, and information sharing to facilitate the timely planning, control, and coordination of the supply chain processes. Thus, IoT technology has played a vital role in operations excellence in supply chain management. This technology has significantly contributed to industrial automation, allowing the integration of industrial sensor networks, radio frequency identification networks, factory control networks, and information management systems. The IoT as a new technology has become more popular since the advent of wireless technology and has attracted the attention of supply chain activists. The IoT has enabled companies to simplify the flow of information and, at all stages of the supply chain, produce significant profits for companies by improving productivity and facilitating the integration of disorganized communications. Through the use of the IoT, companies can make significant gains in productivity at all stages of the supply chain and facilitate in-house and inter-organizational communications [14]. Artificial intelligence also has numerous applications in the supply chain. Areas of impact of artificial intelligence in the supply chain and logistics areas include operational procurement using smart data and chat bots, supply chain planning to forecast supply and demand, warehouse management to optimize inventory, transportation, and faster and more accurate shipping to reduce delivery time and shipping costs and optimally select suppliers through the use of up-to-date data. In addition to artificial intelligence, machine learning is also used in transportation. Organizations now embrace and use machine learning to refine core strategies for issues such as warehouse optimization and day-to-day activities such as availability, costs, inventory, transportation, suppliers, and staff.

Combining these factors has made the IoT and artificial intelligence topics of interest among researchers and industry activists in the supply chain. According to the research work done in IoT and artificial intelligence and supply chain management, it is clear that the research conducted in this field covers a diverse range of topics and different industries. Moreover, some researchers have identified and analyzed the issues emphasized in the theoretical foundations. For example, some have studied the dimensions of using these technologies in the healthcare industry [15,16]. Some studies have pointed to the role of these technologies in creating smart cities [17,18]. In addition, some review studies conducted in this field have specifically studied the IoT in the food and agricultural industries and have considered the results of the application of this technology in the agri-food supply chain [19,20]. Nozari et al. [2], in a paper, provided a framework for IoT-based supply chain and big data analysis. Their research showed how to implement a supply chain emphasizing IoT technologies. In a paper, Arora et al. [21] investigated the effect of integrating artificial intelligence with an IoT-enabled supply chain. In a study, Bamakan et al. [22] also presented an integrated framework based on the Chinese solutions of integrating blockchain, IoT andbig data to evaluate service chain performance. Ghahremani Nahr et al. [23] also provided a framework for using a supply chain emphasizing AIoT in a study. Many studies have pointed to the effects of the simultaneous use of these two technologies and its key role in today’s world. These technologies always play positive roles in today’s world. However, using these technologies can lead to challenges, which will be evaluated in the next section.

## 3. Challenges in AIoT-Based Supply Chain

Technology, as a driving force in accelerating processes in the face of various activities, has played a significant role; one of these important branches of technology is IoT technologies and artificial intelligence, which today plays a significant role in the industry. Nevertheless, these technologies, similar to other emerging technologies, face challenges. In this paper, we try to examine the challenges in the IoT and artificial intelligence within the supply chain as one of the most important parts of manufacturing organizations. Although the concept of smart industries has only been around for a few years with the presence of technologies such as the IoT and the development of the concept and tools of big data analytics, and research has been done in this area, its rapid repetition and upgrading also pose many challenges. In this section, the influencing factors are identified and summarized using a systematic literature review of published articles and studies. These challenges are also assessed using expert opinions, and the most important challenges facing a power supply chain are extracted from IoT technologies and artificial intelligence. In order to analyze the data, experts active in the fields of supply chain and information technology working in FMCG industries were used. The reason for choosing these industries was the availability of this field for researchers. Therefore, seven companies (four food companies and three pharmaceutical companies) were considered as a case study. Since there has not been much proprietary research for the AIoT-based supply chain, the first step was to search for the keywords artificial intelligence and the IoT and their relationship with the supply chain to extract the most important challenges of a smart supply chain. For this purpose, a research question was developed in the first step. In this research, the main questions are: What are the most important key challenges in implementing supply chain based on artificial intelligence, and what category do they have? The statistical population and the search period were determined in the same step. Articles indexed in Scopus and Google Scholar databases were considered to determine the statistical population. It was also considered in light of the background in the field literature.

In the second step, studies related to the research questions were identified. For this purpose, the appropriate keywords for the search must be determined. Keywords for search were: AI and IoT challenges, smart supply chain challenges, IoT-based supply chain implementation challenges, AI-based supply chain implementation challenges, and AI and IoT challenges in the supply chain. In order to extract relevant studies in this field, Scopus and Google Scholar indexing databases were referred to. After searching for keywords in these databases, 278 related document titles were found.

In the third step, the studies that should be reviewed were identified by determining the inclusion criteria. Criteria for inclusion included the English language, quality findings, and the key challenges of using AI and IoT simultaneously in the supply chain. Thus, out of 278 articles published during the review years, 51 non-English items were removed from the review list. After reviewing the titles of the articles and abstracts of the remaining 227 articles, according to the inclusion criteria and the subject, question, and purpose of the research, 24 articles were deleted and 203 articles were considered to review the full content of the article. By reviewing the content of the articles, 20 articles did not have the necessary features for use in the present study and were removed from the study process, and 183 articles with the necessary features for in-depth review and use in this study were identified.

Then, using the modified Delphi method and using the 5-level Likert scale proposed by Azadi et al. [24] with the opinions of experts, 29 challenges as the most important key challenges in implementing the supply chain based on the use of artificial intelligence and IoT were extracted during the studied years. These challenges were then presented to twenty-five experts active in the field of supply chain and information technology in FMCG companies and fifteen experts from the university in this field. Using a questionnaire and a five-level Likert scale, nine major challenges were selected as the most important challenges in this field. These challenges are shown in Table 1.

### 3.1. Cybersecurity

There is a combination of physical and digital systems in an intelligent factory and, accordingly, in an intelligent supply chain that, in addition to generating and maintaining data using IoT technology, uses artificial intelligence for analysis, enabling real-time collaboration, but there is a risk of expanding the level of attack [25,26]. With multiple machines and devices connected to single or multiple networks in intelligent processes, vulnerabilities in any of these devices can open the system to attacks [27]. Companies must anticipate both organizational system vulnerabilities and machine-level operational vulnerabilities. Companies are not always prepared to deal with these security threats, and many rely on their technology and solution providers to do so.

### 3.2. Lack of Trust in AIoT

Trust is often at the level of interpersonal relationships. In today’s intelligent modern life, people’s trust is increasingly systemic. Lack of trust in artificial intelligence and the IoT may slow the development of a strong and intelligent supply chain. However, by reviewing the literature, it can be seen that one of the basic features of systems based on artificial intelligence is their high reliability [28]. However, studies show that there is always concern and distrust of text-based intelligent systems on the IoT. It is seldom claimed that an IoT system works perfectly accurately for any environment, context, and unusual event that the system can experience [29]. Trust means that reliability assessment is highly dependent on accurate knowledge of the context and environment and flexibility to handle unusual events and data. Rarely will such knowledge exist and provide complete flexibility [30].

### 3.3. Connectivity

Wired connectivity is popular in the industry, so pushing the IoT to wireless connectivity could signal a change in network infrastructure design. The use of wireless networks may lead to security concerns [31,32]. In the IoT, digital devices connect and communicate with each other via the Internet, and in small, multi-device networks, the connection is seamless. However, when the IoT is used globally and a number of devices and sensors are connected and communicate with each other, connection problems arise. Moreover, the Internet is not merely a network; it includes heterogeneous networks with cell towers, slow connection, fast connection, proxy servers and firewalls, and different companies with different standards and technologies that can disrupt the connection. Connection is considered one of IoT’s important components because data transfer depends on a good connection [33]. In addition, big data analysis using artificial intelligence and with many connections has its own complexities and is one of the main challenges of a supply chain based on these technologies [34].

### 3.4. Environmental Risks

Businesses are increasingly vulnerable to environmental risks and climate change. A smart supply chain powered by the IoT and artificial intelligence must have highly responsive and agile disaster management systems, such as alerting stakeholders to risk prevention measures and reducing pollution levels [35]. Artificial intelligence and the IoT can be used in the design phase to minimize environmental risks in the business environment as well as product development. Automated drones have been used in various fields, such as detecting and recording environmental hazards, regulating traffic for product distribution, and monitoring environmental pollution [36]. Air quality sensors on online platforms in supply chain processes, especially in manufacturing, can support the measurement of environmental hazards.

### 3.5. Managing Energy

The advantage of IoT devices is that they enable automation. However, a significant amount of energy is required to connect the billions of connected devices to each other. This need for energy, unless managed, can be an obstacle to the full implementation of IoT systems. Technological advances and changes in consumer habits are leading to increased energy demand, and energy producers are now seeking the help of artificial intelligence and IoT to optimize the energy distribution needed [37,38].

### 3.6. Smart Waste

Waste management is one of the main concerns of many industries. This waste is also very high in the production process and has many environmental effects. Waste management from start to finish is one of the key challenges for industrial warehouses worldwide [39]. With the current lifestyle situation in which most food and other items are packaged in plastic or paper packaging, dealing with the waste generated in the manufacturing and industrial sectors is a major concern. Garbage collection must be completed on time, as they require smarter collection [40]. Artificial intelligence and the IoT must address collection, transportation, refining, recycling, and intelligent disposal issues through optimization techniques. Using these technologies, the whole process can be centrally monitored and, as a result, services are provided in a smart supply chain [41].

### 3.7. Managing Transportation

In today’s world, transportation is one of the most critical sectors of the economy. Although transportation has greatly improved modern life, many problems in this area are still unresolved. Current modes of transportation are heavily dependent on crude oil products such as oil, diesel, and so on. Electric vehicles are a good alternative to combat greenhouse gas emissions. Electric vehicles use batteries and electric motors to generate the power needed to drive [42]. If existing fuel stations are upgraded to hybrid modes that offer petroleum products as well as electronic charging stations, the charging requirements of electric vehicles can be met. To avoid traffic jams, urban design is important to minimize the need for daily public transportation, filling this gap and reducing residents’ travel time [43]. Machine learning methods that are an essential part of artificial intelligence must be sufficiently capable of analyzing past data from public and private transportation activities to address the root causes of frequent congestion or most accidents and disruption and distribution. Preventive measures are needed to address them [44].

### 3.8. Lack of Proper Infrastructure

The smart industries, and consequently the smart supply chain, need the latest highly advanced infrastructure, and every piece of equipment must be connected to the Internet to monitor it. In the smart supply chain, IoT-connected devices collect data from physical media to optimize decisions to improve all supply chain processes from supply to distribution. Creating an information service to support information is one of the tasks of smart companies to create a smart and robust supply chain [45,46]. Digital supply chain services must have the infrastructure necessary to build cyber and physical systems. Smart factory services must have the infrastructure components necessary to complete the technological operation of the product in automatic mode with cyber equipment [47]. The smart supply chain infrastructure must have the infrastructure necessary for personnel to control the item during remote operation [48].

### 3.9. Lack of Professionals

To successfully implement new technology and maintain operations, a company must have a workforce with “digital skills”—people must understand both the production processes and the digital tools that support those processes. In the absence of forces familiar with digital developments and the IoT or artificial intelligence, in addition to slowing down all progressive activities, there is organizational resistance to implementing AIoT-based intelligent processes in organizations [49]. Thus, training in digital tools and skills (which is important in today’s world but vital to the future) must be provided. Incorporating concepts related to cybersecurity, digital infrastructure, artificial intelligence, big data, storage, and computing needs into the educational content of the actors involved in supply chain activities must be implemented to continue to succeed in the world to come [50].

## 4. Research Methodology

This research is applied in terms of purpose and is descriptive in nature. In this study, in the first stage, library studies were used to prepare a list of challenges facing an AIoT-based supply chain or the simultaneous use of IoT and artificial intelligence technologies, and then the most important challenges were selected using the opinions of experts. Therefore, experts’ knowledge was used in the challenge ranking stage. Since the information required to prioritize challenges is based on the knowledge of industrial and academic experts, and on the other hand, the number of people with sufficient knowledge and experience in information management in the supply chain is limited, we chose to select experts using the sampling method. In order to analyze the data in this research, industrial and academic experts have been used. Therefore, among the people active in the field of information technology and supply chain in FMCG industries, 30 specialists, who had more than 5 years of experience and had sufficient knowledge in the field of artificial intelligence and IoT and worked in the field of the supply chain of these important industries, were considered as industrial experts. Moreover, 15 university specialists whose specialized background in the university was related to research and had academic and industrial experience were considered as academic experts. The computational results confirmed the consistency between the experts’ opinions, which indicates the correct choice and alignment of the experts’ opinions. Therefore, judicial and purposeful research has been used. This research was conducted in the fall of 2021, and the FMCG industry in Iran was selected as a case study. The reason for choosing this industry was the availability of companies as well as the presence of knowledgeable experts in the field of digital transformation in these companies. Considering that the present research is about extracting knowledge from experts’ minds, most of it is in the field of qualitative studies. Solving methods in this research are structural modelling and fuzzy decision-making method DEMATEL to investigate the internal impact of challenges. In addition, a group fuzzy priority planning method is used to understand the importance of challenges and rank them. This method is a practical and approved method in the subject literature and has a short run time and can be implemented in the fastest possible time. In order to converge the answers for the reliability of the questionnaires, we tried to monitor the scatter of experts’ answers visually. The evaluation model of this research is summarized as follows.

Step 1: Identify and select the most important challenges of implementing an AIoT-based supply chain. In the present study, we first tried to extract the challenges faced by an intelligent supply chain powered by both IoT and artificial intelligence technologies by reviewing the literature among Google Scholar, Scopus, and Web of Science databases. Then, using the opinions of experts active in the field of supply chain and information technology working in FMCG companies (due to the authors’ access to these experts), three categories (security, environmental, and managerial), as the main challenges in implementing a supply chain based on AIoT, were selected.

Step 2: Create a hierarchical decision structure. At this stage, the hierarchical structure of the decision was determined using the target levels, criteria, and options (Table 1).

Step 3: Determine the internal relationships between the challenges. At this stage, using the fuzzy DEMATEL method, an attempt is made to examine how the challenges are addressed.

Step 4: Determine the weight and priority of each of the challenges facing an AIoT-based supply chain. Finally, in this stage, using the Mikhailov fuzzy nonlinear mathematical model [51], which is based on pairwise comparisons in the AHP method, the existing challenges are prioritized and the most important challenges ahead are selected.

The following will examine the decision-making methods used in this research.

### 4.1. Fuzzy DEMATEL Method

Fuzzy DEMATEL is a way to identify the pattern of causal relationships between decision criteria with a fuzzy inference approach. This method is an extension of the traditional DEMATEL method using fuzzy logic. For the first time, [52] used the fuzzy approach DEMATEL technique in a paper entitled “Developing global managers’ competencies using the fuzzy DEMATEL method”. In fact, the fuzzy approach is used to deal with the uncertainty and ambiguity in the verbal expressions of the respondents.

Therefore, to perform the calculations of the DEMATEL method with the fuzzy approach in the first fuzzy stage, a suitable linguistic spectrum must be used for data collection. A variety of ranges are suggested based on the conventional DEMATEL rating scale. The fuzzy language scale is given in Table 2.

Triangular fuzzy numbers equivalent to the DEMATEL spectrum are shown in Figure 1.

Several models have been proposed to perform fuzzy DEMATEL calculations. Defuzzification methods have heavily influenced a common pattern used.

The fuzzy DEMATEL implementation algorithm is as follows.

Step 1: Calculate the direct relation matrix

After gathering the experts’ points of view, the fuzzy direct communication matrix $\tilde{X}$ is formed. The simple fuzzy mean method is used to aggregate the opinions of experts. If n experts exist and each direct fuzzy matrix object is represented by ${\tilde{X}}_{ij}$, then ${\tilde{X}}_{ij}$ is calculated as follows:(1)$${\tilde{X}}_{ij}=(\frac{\sum l_{ij}}{n},\frac{\sum m_{ij}}{n},\frac{\sum u_{ij}}{n})$$

Step 2: Normalize the direct relation matrix

To normalize the values, $\sum u_{ij}$ per row must be calculated. By dividing the matrix $\tilde{X}$ by the maximum values, the $\sum u_{ij}$ values of the fuzzy normal matrix are obtained:(2)$$k=\max\left(\sum_{j=1}^{n}u_{ij}\right)\;\tilde{N}=\frac{1}{k}\times \tilde{X}$$

Step 3: Calculate the complete relation matrix

The relation $N\times {\left(I-N\right)}^{-1}$ is used to calculate the complete correlation matrix. In fuzzy DEMATEL, the normal fuzzy matrix is subdivided into three definite matrices:(3)$$N_{u}=\left[\begin{array}{cccc} 0 & u_{12} & \cdots & u_{1n}\\ u_{21} & 0 & \cdots & u_{2n}\\ \vdots & \vdots & \ddots & \\ u_{n1} & u_{n2} & \cdots & 0 \end{array}\right],N_{m}=\left[\begin{array}{cccc} 0 & m_{12} & \cdots & m_{1n}\\ m_{21} & 0 & \cdots & m_{2n}\\ \vdots & \vdots & \ddots & \\ m_{n1} & m_{n2} & \cdots & 0 \end{array}\right],\;N_{l}=\left[\begin{array}{cccc} 0 & l_{12} & \cdots & l_{1n}\\ l_{21} & 0 & \cdots & l_{2n}\\ \vdots & \vdots & \ddots & \\ l_{n1} & l_{n2} & \cdots & 0 \end{array}\right]$$

Then, the identity matrix $I_{n\times n}$ is formed and the following operations are performed:(4)$$T_{l}=N_{l}\times {\left(I-N_{l}\right)}^{-1},\;T_{m}=N_{m}\times {\left(I-N_{m}\right)}^{-1},\;T_{u}=N_{u}\times {\left(I-N_{u}\right)}^{-1}$$
(5)$${\tilde{t}}_{ij}=(t_{ij}^{l},t_{ij}^{m},t_{ij}^{u})$$

We define r and c as two matrices n×1, which represent the sum of the rows and columns of the complete relation matrix.
(6)$$r={\left[r_{i}\right]}_{n\times 1}={\left(\sum_{j=1}^{n}t_{ij}\right)}_{n\times 1}$$
(7)$$c={\left[c_{j}\right]}^{\prime }{}_{1\times n}={\left(\sum_{i=1}^{n}t_{ij}\right)}^{\prime }{}_{1\times n}$$

$r_{i}$ is equal to the sum of the ith row of the total relation matrix T. Thus, $r_{i}$ represents the effect of the total factor i on other factors. This effect includes direct and indirect effects. $c_{j}$ is equal to the sum of jth columns of the matrix of the total relation T. Thus, $c_{j}$ represents the overall effect that factor j has received from other factors. This effect includes direct and indirect effects. Therefore, when j=i, then ($r_{i}+c_{i}$) is equal to the total effect applied and received by factor i. In other words, ($r_{i}+c_{i}$) indicates the degree of importance of factor i in the system. ($r_{i}-c_{i}$) also represents the net effect that factor i exerts on the entire system. When ($r_{i}-c_{i}$) is a positive value, it means that factor i is an influential factor in the system as a whole, and when ($r_{i}-c_{i}$) is a negative value, it means that factor i is an influential factor in the whole system [53].

Step 4: Determine the threshold value and obtain the effect relationship map

In many studies, in order to show the structural relationship between the factors, while maintaining the complexity of the system in a controllable way, it is necessary to determine the value of the p thresholds in such a way that only the negligible effects in the T matrix are filtered. Only effects in the T matrix that are larger than the threshold value should be selected and displayed in the impact relationship map (IRM) or causal relationship map diagram.

### 4.2. Nonlinear Fuzzy Prioritization

This research uses a fuzzy nonlinear prioritization method to measure the weight and rank of challenges in implementing an AIoT-based smart supply chain. Since the fuzzy weight determination methods use a pairwise comparison matrix, and inspired by the definitive hierarchical analysis method, the reciprocal matrix is explained, which leads to problems. In addition, sometimes, decision makers may or may not want to make all the comparisons, so the Mikhailov nonlinear method is used in this study. The steps for using this method are as follows.

Step 1. Drawing the hierarchical structure. In this step, the hierarchical structure of the decision is drawn using the target levels of criteria and options. This structure is presented in Table 1.

Step 2. Formation of fuzzy judgment matrix. Agreed fuzzy judgment matrices are formed based on decision makers’ opinions. Therefore, it is necessary to use fuzzy numbers in explaining people’s preferences and polls, which is important in this study. In this method, fuzzy language scales will be used to obtain expert opinions. These linguistic scales for matrices of pairwise comparisons and their fuzzy equations are given in Table 3.

Step 3. Formulation and solution of the model. In this method, even fuzzy comparisons are assumed to be fuzzy triangular numbers. The definite weight vector (priority) $w=\left(w_{1},\;w_{2},\ldots ,\;w_{n}\right)$ is extracted in such a way that the priority rate is approximately within the range of the initial fuzzy judgments. In other words, the weights are determined to establish the following relation.
(8)$$l_{ij}\leq \frac{w_{i}}{w_{j}}\leq u_{ij}$$

Each definite weight vector (w) holds with a degree in the above fuzzy inequalities, which can be measured by the linear membership function of the following relation (in terms of unknown rate):(9)$$\mu_{ij}\left(\frac{w_{i}}{w_{j}}\right)=\left\{\begin{array}{c} \frac{(w_{i}/w_{j})-l_{ij}}{m_{ij}-l_{ij}}\;\;\;\frac{w_{i}}{w_{j}}\leq m_{ij}\\ \frac{u_{ij}-(w_{i}/w_{j})}{u_{ij}-m_{ij}}\;\;\;\frac{w_{i}}{w_{j}}\leq m_{ij} \end{array}\right.$$

Given the specific form of membership functions, the fuzzy prioritization problem becomes a nonlinear optimization problem as follows.
(10)$$\begin{array}{l} \max\;\lambda \\ Subject\;to:\\ (m_{ij}-l_{ij})\lambda w_{j}-w_{i}+l_{ij}w_{j}\leq 0\\ (u_{ij}-m_{ij})\lambda w_{j}+w_{i}-u_{ij}w_{j}\leq 0\\ i=1,2,\ldots ,n-1,\;\;j=2,3,\ldots ,n,\;\;j>i,\\ \sum_{k=1}^{n}w_{k}=1\;\;w_{k}>0,\;\;k=1,2,\ldots ,n \end{array}$$

Due to the model’s nonlinearity, it is impossible to solve it by the simplex method and it must be solved using appropriate quantitative and software methods. Positive optimal values for the index indicate that all weight ratios are entirely true in the initial judgment, but if the index is negative, it can be seen that the fuzzy judgments are strongly inconsistent and the weight ratios are almost true in these judgments.

## 5. Research Findings

### 5.1. Internal Relationship of Supply Chain Implementation Challenges Based on AIoT

In this research, we use the fuzzy DEMATEL method to find the effect of factors on each other. For this purpose, for accurate analysis, questionnaires were sent to forty-five experts. Thirty of these specialists were active in FMCG companies and in the supply chain and information technology departments, and fifteen questionnaires were sent to academic specialists. The experts had industrial and academic backgrounds in the field under study and were selected according to the access of the researchers. The experts were asked to express their views on the extent of the internal effects of these challenges based on linguistic variables. Of these questionnaires, forty questionnaires were completed and received. As a result, the fuzzy direct relation matrix was formed for the performance indicators, as presented in Table 4.

In Table 5, the general fuzzy relation matrix for AIoT-based supply chain implementation challenges is shown.

The sum of the elements in the columns and rows of the whole matrix was calculated for the implementation challenges to present the relational map. These values are called effective vectors (*R*) and effective vectors (*D*). The results are shown in Table 6.

Table 6 shows the results of internal impact calculations of the challenges of implementing an intelligent supply chain based on the combined use of artificial intelligence and the IoT. As shown in the table, the lack of proper infrastructure is one of the most influential challenges affecting others. The effective internal relationships of the challenges are shown in Figure 2.

### 5.2. Ranking of Implementation Challenges Using Nonlinear Fuzzy Prioritization Method (Mikhailov)

The steps related to evaluating and ranking the challenges of implementing an AIoT-based supply chain in this study are divided into two main parts:

Determining the matrix of pairwise comparisons based on the integration of experts;Applying mathematical modeling in ranking and obtaining the weights of factors in the research model.

In order to evaluate and prioritize the challenges in this study, fuzzy questionnaires using language variables were sent to 30 experts active in the FMCG industry, as well as 15 academic experts. In total, 40 questionnaires were completed and received. These pairwise comparison tables are shown as Table 7, Table 8 and Table 9. These tables have been used for calculations in the Mikhailov method.

After this step and reviewing the experts’ opinions, we used the data of these matrices obtained in mathematical modeling for ranking. For this purpose, fuzzy values were placed in the nonlinear mathematical model. Since the model was nonlinear, Lingo software was used to solve the model. The results showed the weight of each of the factors, and they are shown in Table 10, Table 11 and Table 12.

Next, using the information obtained from Table 10, Table 11 and Table 12, we obtained the normal weight for the challenge group. This normalized weight is shown in Table 13.

Considering the normal weights obtained in Table 13, it can be seen that the lack of proper infrastructure is always one of the most important challenges in implementing an intelligent supply chain based on artificial intelligence and IoT technologies. In addition, and as it is possible, security challenges are also one of the most important challenges of these smart supply chains. Therefore, senior managers of organizations need to pay special attention to these points to create an intelligent framework in the supply chain.

## 6. Discussion and Conclusions

Industrial intelligence means using control devices such as computers instead of using humans to guide and control industrial machinery during the production process, which aims to reduce the need for human intervention and increase the speed of industrial production. It is considered one of the main infrastructures in all industries, equipment, and devices for industrial control and intelligence, so it is important to consider the extent to which the intelligence of industries can be effective in transforming a country’s industry.

A supply chain is one of the areas affected by the Fourth Industrial Revolution and digital technologies such as the IoT, artificial intelligence, advanced robotics, and big data analytics in today’s world. The concept of the intelligent supply chain is important to provide sustainable conditions in all chain processes. However, achieving stability is difficult due to the involvement of multiple factors. Artificial intelligence and IoT technologies can change challenges and provide compelling solutions to problems that actors involved in industrial processes understand. For this reason, this study was conducted to identify and analyze the impact of important challenges that hinder the adoption of artificial intelligence and the IoT in developing a sustainable, intelligent supply chain. Therefore, this study first tried to extract the most important challenges in implementing the intelligent supply chain affected by the IoT and artificial intelligence using a literature review. Then, using experts’ opinions in FMCG industries, the most important of these challenges were selected as a case study. Then, using the fuzzy DEMATEL method, the internal effects of these challenges on each other were determined. The absence of appropriate infrastructure was identified as one of the most influential and environmental hazards as one of the most compelling challenges. Then, using nonlinear ranking analysis and based on matrix comparison matrices, the importance of each of the implementation challenges was examined.

The results showed that the challenges related to the lack of proper infrastructure are among the most important challenges of implementing a system based on digital transformation needs. This part can be one of the basic requirements for implementation in any other technology. As a result, many companies do not take the path of implementation without the appropriate infrastructure (technological and technical, as well as the lack of necessary preparation in the organization to accept technology). This infrastructure is one of the components of organizational maturity that confirms the high ranking of this option. Security challenges are in second place; given the presence of Internet and network technology, it can be seen how important issues related to cybersecurity can be. Cybersecurity and privacy risks are a major concern for researchers and security professionals. This poses significant challenges for many business organizations. Common cybersecurity attacks have shown the vulnerability of IoT technologies. This vulnerability is simply due to the interconnectedness of networks in the Internet of Things, which provides access through anonymous and unreliable Internet connections, which requires new security solutions. None of the known challenges have a significant impact on IoT compatibilities, such as security and privacy. However, many users often do not have the necessary confirmation of security effects until a problem occurs, and this can lead to many damages, such as the loss of important information. In the next stage, the lack of sufficient knowledge of the most important challenges in implementing a digital supply chain based on the IoT and artificial intelligence should be considered. The organization must take steps to increase public knowledge about digital developments among the general staff of organizations to increase the organization’s readiness to accept technologies and increase confidence in the use of these technologies in industrial cycles. Therefore, it can be seen that managing these challenges is one of the most critical tasks of technology development managers in organizations and can increase the agility and stability of the supply chain to an acceptable level. As highlighted in this study, it should be noted that, in addition to the existence and implementation of technological infrastructure in various industries, the presence of capable and expert human resources is necessary to understand better the concepts related to the effects of digital transformation, and thus remove these limitations and train experts. This can increase the accuracy of research to the latest technologies in the world. The presence of IoT technology in the supply chain of other industries as a case study can also provide a more powerful model and framework. Therefore, in addition to examining the presence of IoT technology in the supply chain of other industries (to provide more powerful and accurate prioritization), the challenges of the presence of many transformational technologies, such as blockchain, machine learning, etc., can be examined in future research.

## Figures and Tables

**Figure 1 sensors-22-02931-f001:**
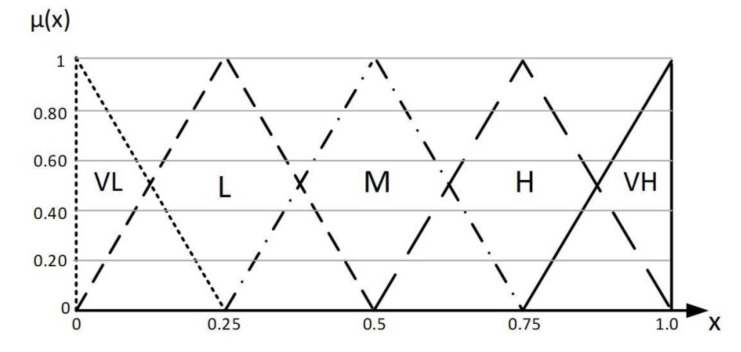
Membership function of triangular fuzzy numbers.

**Figure 2 sensors-22-02931-f002:**
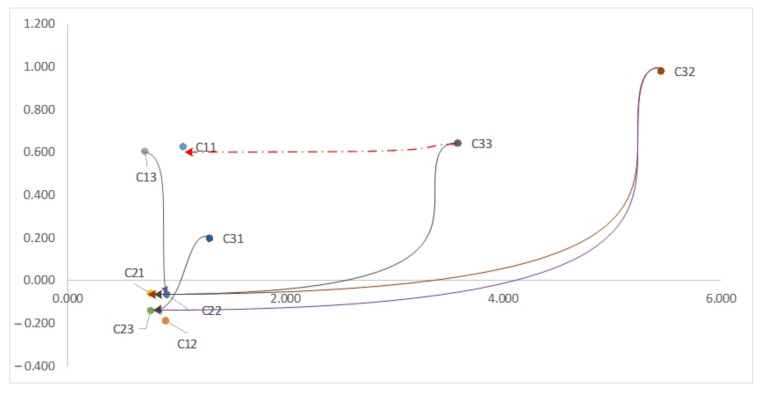
Internal impacts of AIoT-based supply chain implementation challenges.

**Table 1 sensors-22-02931-t001:** Units for magnetic properties.

Code	Challenges	Category
C11	Cybersecurity	Security (C1)
C12	Lack of Trust in AIoT	
C13	Connectivity	
C21	Environmental Risks	Environmental (C2)
C22	Managing Energy	
C23	Smart Waste	
C31	Managing Transportation	Managerial (C3)
C32	Lack of Proper Infrastructure	
C33	Lack of Professionals	

**Table 2 sensors-22-02931-t002:** Fuzzy linguistic scale.

Linguistic Terms	Triangular Fuzzy Number
Very low (VL)	(0, 0, 0.25)
Low (L)	(0, 0.25, 0.5)
Medium (M)	(0.25, 0.5, 0.75)
High (H)	(0.5, 0.75, 1)
Very high (VH)	(0.75, 1, 1)

**Table 3 sensors-22-02931-t003:** Linguistic scale for the pairwise comparison matrix.

Linguistic Values for Pairwise Comparisons	Triangular Fuzzy Scales
Very low (VL)	(1, 2, 3)
Low (L)	(2, 3, 4)
Medium (M)	(3, 4, 5)
High (H)	(4, 5, 6)
Very high (VH)	(5, 6, 7)

**Table 4 sensors-22-02931-t004:** Fuzzy direct relationship matrix between AIoT-based supply chain implementation challenges (summarized).

	C11			C12			C…	C32			C33		
	*L*	*M*	*U*	*L*	*M*	*U*	*…*	*L*	*M*	*U*	*L*	*M*	*U*
C11	0	0	0	0.75	0.45	0.25	…	0.55	0.35	0.1	0.95	0.35	0.17
C12	0.9	0.85	0.38	0	0	0	…	0.75	0.5	0.3	0.8	0.32	0.25
…	…	…	…	…	…	…	…	…	…	…	…	…	…
C32	0.75	0.71	0.41	0.9	0.81	0.25	…	0	0	0	0.9	0.84	0.42
C33	0.7	0.6	0.3	0.85	0.55	0.45	…	0.91	0.84	0.41	0	0	0

**Table 5 sensors-22-02931-t005:** Total fuzzy relation matrix for AIoT-based supply chain implementation challenges (summarized).

	C11			C12			C…	C32			C33		
	*L*	*M*	*U*	*L*	*M*	*U*	*…*	*L*	*M*	*U*	*L*	*M*	*U*
C11	0.12	0.11	0.05	0.21	0.2	0.06	…	0.22	0.11	0.09	0.21	0.11	0.08
C12	0.21	0.14	0.04	0.21	0.11	0.038	…	0.2	0.12	0.07	0.22	0.13	0.06
…	…	…	…	…	…	…	…	…	…	…	…	…	…
C32	0.21	0.13	0.1	0.24	0.14	0.11	…	0.14	0.11	0.03	0.17	0.15	0.08
C33	0.24	0.15	0.1	0.21	0.12	0.1	…	0.2	0.17	0.12	0.19	0.16	0.07

**Table 6 sensors-22-02931-t006:** Results of calculating the internal effects of implementation challenges.

Challenges	D	R	D+R	D−R
Cybersecurity	0.841	0.214	1.055	0.627
Lack of Trust in AIoT	0.354	0.541	0.895	−0.187
Connectivity	0.657	0.0487	0.706	0.608
Environmental Risks	0.354	0.411	0.765	−0.057
Managing Energy	0.421	0.487	0.908	−0.066
Smart Waste	0.311	0.452	0.763	−0.141
Managing Transportation	0.751	0.554	1.305	0.197
Lack of Proper Infrastructure	3.214	2.234	5.448	0.980
Lack of Professionals	2.114	1.471	3.585	0.643

**Table 7 sensors-22-02931-t007:** Parallel comparison matrix of security challenges.

	C11			C12			C13		
	W1			W2			W3		
W1	-	-	-	-	-	-	-	-	-
W2	2.7	2.9	6.5	-	-	-	-	-	-
W3	2.6	2.4	3.1	1.1	1.3	4.2	-	-	-

**Table 8 sensors-22-02931-t008:** Parallel comparison matrix of challenges in the environmental category.

	C21			C22			C23		
	W1			W2			W3		
W1	-	-	-	-	-	-	-	-	-
W2	2.5	2.8	4.1	-	-	-	-	-	-
W3	3.1	3.5	4.0	2.1	2.5	5.1	-	-	-

**Table 9 sensors-22-02931-t009:** Parallel comparison matrix of challenges in the management category.

	C21			C22			C23		
	W1			W2			W3		
W1	-	-	-	-	-	-	-	-	-
W2	3.1	3.4	5.7	-	-	-	-	-	-
W3	3.4	4.1	4.2	3.1	3.5	6.0	-	-	-

**Table 10 sensors-22-02931-t010:** Ranking AIoT-based supply chain implementation challenges in the security category.

Challenges	Code	Weight	Rank
Cybersecurity	W1	0.536274	1
Lack of Trust in AIoT	W2	0.413259	2
Connectivity	W3	0.051289	3

**Table 11 sensors-22-02931-t011:** Ranking AIoT-based supply chain implementation challenges in the environmental category.

Challenges	Code	Weight	Rank
Environmental Risks	W1	0.457404	1
Managing Energy	W2	0.257241	3
Smart Waste	W3	0.284401	2

**Table 12 sensors-22-02931-t012:** Ranking AIoT-based supply chain implementation challenges in the management category.

Challenges	Code	Weight	Rank
Managing Transportation	W1	0.095241	3
Lack of Proper Infrastructure	W2	0.546234	1
Lack of Professionals	W3	0.362154	2

**Table 13 sensors-22-02931-t013:** Normal weight and ranking of AIoT-based supply chain implementation challenges in FMCG industries.

Category	Challenges	Weight	Normal Weight	Rank
Security	Cybersecurity	0.536274	0.177583	2
	Lack of Trust in AIoT	0.413259	0.136847	4
	Connectivity	0.051289	0.016984	9
Environmental	Environmental Risks	0.457404	0.11625	5
	Managing Energy	0.257241	0.065378	7
	Smart Waste	0.284401	0.072281	6
Managerial	Managing Transportation	0.095241	0.039824	8
	Lack of Proper Infrastructure	0.546234	0.2284	1
	Lack of Professionals	0.362154	0.151429	3

## Data Availability

Not applicable.

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
