# Peer review of "Analysis of the Challenges of Artificial Intelligence of Things (AIoT) for the Smart Supply Chain (Case Study: FMCG Industries)"

_sensors, 2022, doi:10.3390/s22082931_

Round 1
Reviewer 1 Report
After reading the manuscript, it should be have sufficient modifications before publication. Detailed suggestions are shown as follow.
(1) Page 2, Line 55 You explained Internet of Things is IoT, please use abbreviation. Please check the whole paper to achieve consistency
(2) Page 2, Line 71-73 You cannot do such strong claim without reference
(3) Page 2, Line 78 What is the meaning of FMCG?
(4) Page 2, Line 79-80 This sentence is not complete
(5) Page 2, Line 88 What is the meaning of IT?
(6) Page 3, Line 143 It should be 2.2 AI and IoT-based supply chain. Please make changes at the whole manuscript
(7) Page 4 Line 147 Where are the references to support this claim?
(8) Page 4 Line 151-154 Where is the page number?
(10) Page 4 Line 182 If you are using many studies in here, why only two references support your claim? Please check the whole paper to ensure you do not have this kind of problem
(11) Page 5 Line 216-225 I believe that you did not perform a systematic literature review to search relevant challenges. If you used systematic literature review, please provide keywords you used, exclude/include criteria for searching papers, etc.
(12) Page 7 Line 7 Some references you used numbers, please check the whole paper to keep consistency
(13) Page 8, Line 342 Are you talking about purposive sampling? How many experts are you selected? Detailed information? if there is a conflict between experts, how do you deal with the situation?
(14) Page 8, Line 347-350 There are different multi-criteria decision making methods can be used, for example, AHP, TISM, ISM, ANP, TOPSIS and others. Why you only choose these two methods? Justifications?
(15) Page 12 Line 464 What is the meaning of final effect?
(16) Page 12 Line 466 45 experts? Why so many experts? Is there conflict opinion between two experts? How do you deal with this situation? Detailed criteria to choose experts? Which software you used to process data analysis?
(15) Page 15 Line 534 Where is the discussion section? Without proper discussion, how do you evaluate your contributions?
(16) Page 15 Line 553-555 I think you do not repeat this justification in conclusion.
(17) Page 15, Line 562-563 Where are the limitations and future research directions? Where are the implications for research and management practices?
Author Response
Response to Reviewer No. 1
- Page 2, Line 55 You explained Internet of Things is IoT, please use abbreviation. Please check the whole paper to achieve consistency
Thanks to the comments of the dear reviewer, all the words Internet of Things were replaced by abbreviations.
- Page 2, Line 71-73 You cannot do such strong claim without reference
Thanks to the reviewer, this section has been modified.
- Page 2, Line 78 What is the meaning of FMCG?
Thanks to the reviewer, this definition was added to the article.
- Page 2, Line 79-80 This sentence is not complete
Thanks to the reviewer, this section has been modified.
- Page 2, Line 88 What is the meaning of IT?
Thanks to the reviewer, this definition was added to the article.
- Page 3, Line 143 It should be 2.2 AI and IoT-based supply chain. Please make changes at the whole manuscript
Thanks to the reviewer, this section has been modified.
- Page 4 Line 147 Where are the references to support this claim?
Thanks for the remark of the reviewer, this reference was added.
- Page 4 Line 151-154 Where is the page number?
Thanks to the reviewer, this definition is given on page 4 of the reference article.
- Page 4 Line 182 If you are using many studies in here, why only two references support your claim? Please check the whole paper to ensure you do not have this kind of problem
Thanks to the remark of the reviewer, this sentence was corrected and other parts of the article were also checked.
- Page 5 Line 216-225 I believe that you did not perform a systematic literature review to search relevant challenges. If you used systematic literature review, please provide keywords you used, exclude/include criteria for searching papers, etc.
Thanks to the reviewer, this section was added to the article.
- Page 7 Line 7 Some references you used numbers, please check the whole paper to keep consistency
Thanks, the references have been corrected.
- Page 8, Line 342 Are you talking about purposive sampling? How many experts are you selected? Detailed information? if there is a conflict between experts, how do you deal with the situation?
Thanks to the esteemed reviewer, additional explanations were added.
- Page 8, Line 347-350 There are different multi-criteria decision making methods can be used, for example, AHP, TISM, ISM, ANP, TOPSIS and others. Why you only choose these two methods? Justifications?
Our goal in this paper is to understand the internal relationships as well as the importance of the key challenges of implementing an intelligent supply chain. Therefore, we wanted to use the tools that have been used in the literature and can lead us to our goal. In order to examine the internal relations, we used the DEMATEL method, which is one of the best decision-making tools. For prioritization, we used the Mikhailov method, which in addition to being a practical method, does not require a high run time, and its strength is emphasized in the literature. It also shows the compatibility of the questionnaires in a simple way that It is very important to understand the quality of research.
- Page 12 Line 464 What is the meaning of final effect?
Thanks to the reviewer, this sentence was corrected
- Page 12 Line 466 45 experts? Why so many experts? Is there conflict opinion between two experts? How do you deal with this situation? Detailed criteria to choose experts? Which software you used to process data analysis?
Experts have industrial and academic backgrounds in the field under study and were selected according to the access of researchers. The incompatibility rate of the questionnaires was less than 0.1, which showed that the opinions of the experts were in line, and therefore we did not encounter any problems in the analysis of the questionnaires. Excel and MATLAB software were also used for the analysis.
- Page 15 Line 534 Where is the discussion section? Without proper discussion, how do you evaluate your contributions?
Thanks to the reviewer, this section was added
- Page 15 Line 553-555 I think you do not repeat this justification in conclusion.
Based on the opinion of the esteemed reviewer, the sentence was deleted
- Page 15, Line 562-563 Where are the limitations and future research directions? Where are the implications for research and management practices?
Thanks to the reviewer, this section was added
Reviewer 2 Report
The proposed study is thought to have suggested a method that can be effectively used to make important policy decisions.
The FMCG industry in Iran was selected and analyzed as a case study, and the fuzzy decision-making method DEMATEL was used. The overall research method and results are thought to be appropriate, but there is a problem with the lack of objectivity for expert knowledge.
In order for the research to be accepted, it is thought that a method to show the objectivity of expert knowledge is needed.
Author Response
Response to Reviewer No. 2
The proposed study is thought to have suggested a method that can be effectively used to make important policy decisions.
The FMCG industry in Iran was selected and analyzed as a case study, and the fuzzy decision-making method DEMATEL was used. The overall research method and results are thought to be appropriate, but there is a problem with the lack of objectivity for expert knowledge.
In order for the research to be accepted, it is thought that a method to show the objectivity of expert knowledge is needed.
Thanks to the careful consideration of the esteemed reviewer, as mentioned in the article, in order to select experts from among those who have work experience as well as academics in the field under study, people were selected who have experience and in the case under study. Have had. At least 5 years of work experience in the field of study for industrial experts as well as research records for academic experts have been the criteria for selecting experts.
Round 2
Reviewer 1 Report
The author(s) successfully tackled all comments. I suggest to accept it at present form.